# Development of a Human Estrogen Receptor Dimerization Assay for the Estrogenic Endocrine-Disrupting Chemicals Using Bioluminescence Resonance Energy Transfer

**DOI:** 10.3390/ijerph18168875

**Published:** 2021-08-23

**Authors:** Hye Mi Kim, Hyeyeong Seo, Yooheon Park, Hee-Seok Lee, Seok-Hee Lee, Kwang Suk Ko

**Affiliations:** 1Department of Nutritional Science and Food Management, Ewha Womans University, Seoul 03760, Korea; gpaldlt@gmail.com; 2Department of Integrated Biomedical and Life Science, Korea University, Seoul 02841, Korea; hishyung20@gmail.com; 3Department of Food Science and Biotechnology, Dongguk University, Goyang 10326, Korea; ypark@dongguk.edu; 4Department of Food Science and Technology, Chung-Ang University, Anseong 17546, Korea; hslee0515@cau.ac.kr

**Keywords:** estrogenic endocrine-disrupting chemical, estrogen receptor, bioluminescence resonance energy transfer, risk assessment

## Abstract

Endocrine-disrupting chemicals (EDCs) are found in food and various other substances, including pesticides and plastics. EDCs are easily absorbed into the body and have the ability to mimic or block hormone function. The radioligand binding assay based on the estrogen receptors binding affinity is widely used to detect estrogenic EDCs but is limited to radioactive substances and requires specific conditions. As an alternative, we developed a human cell-based dimerization assay for detecting EDC-mediated ER-alpha (ERα) dimerization using bioluminescence resonance energy transfer (BRET). The resultant novel BRET-based on the ERα dimerization assay was used to identify the binding affinity of 17β-estradiol (E2), 17α-estradiol, corticosterone, diethylhexyl phthalate, bisphenol A, and 4-nonylphenol with ERα by measuring the corresponding BRET signals. Consequently, the BRET signals from five chemicals except corticosterone showed a dose-dependent sigmoidal curve for ERα, and these chemicals were suggested as positive chemicals for ERα. In contrast, corticosterone, which induced a BRET signal comparable to that of the vehicle control, was suggested as a negative chemical for ERα. Therefore, these results were consistent with the results of the existing binding assay for ERα and suggested that a novel BRET system can provide information about EDCs-mediated dimerization to ERα.

## 1. Introduction

Endocrine-disrupting chemicals (EDCs) interfere with the functions of the endocrine system by mimicking or blocking hormone function in humans and animals. Absorption of EDCs in vivo can inhibit or stimulate the production of hormones and their metabolisms to increase the risk of diseases, such as reproductive disorders and hormone-related cancer [1,2]. A considerable number of EDCs are widely found in everyday products, such as industrial solvents/byproducts, agricultural pesticides, plastics, and other manufactured products, and also include natural chemicals, such as phytoestrogen [3,4]. These contaminants may enter the food chain by direct contact with food components [5,6]. For instance, phenolic EDCs from industrial sewage plant flow in the aquatic environment and bisphenol A, another kind of EDC, are contained in plastics used for food packing materials or beverage containers. Exposure and accumulation of bisphenol A in the body can cause changes in body weight and fat mass or increasing diabetes or cardiovascular disease [7,8,9]. Among EDCs found everywhere in human life, including foods and food ingredients, are the chemicals that disrupt the physiological functions of estrogen, which are called estrogenic EDCs [10]. These estrogenic EDCs were associated with behavioral changes, reproductive disorders, and cancer development in both vertebrates and invertebrates by mimicking estrogenic functions [11]. Therefore, to detect or evaluate estrogenic EDCs that interfere with normal estrogen function is necessary because estrogenic EDCs can be released into the environment and are subsequently exposed to the human body [12].

Various methods exist to evaluate estrogenic EDCs. These assays are based on the molecular and cellular mechanisms of estrogen action, such as reporter gene assay, yeast two-hybrid assay, transcription assay, protein assay, cell assay, animal tests, and ligand-binding assay [13]. Among the in vitro methods of detecting EDCs, the performance-based test guideline for human recombinant estrogen receptor-alpha (ERα) using binding assay (OECD TG 493) recommended by the Organisation for Economic Co-operation and Development (OECD) is the most popular method for the initial screening step and is regarded as the most convenient approach [14]. This radioligand binding assay [15] measures the ability of a radiolabeled ligand ([^3^H]17β-estradiol) to bind with ER-alpha (ERα) when in the presence of competitor chemicals. The main advantages of this method are its sensitivity and specificity. However, its disadvantages include relatively long read times, high cost, health risk, requirement for specific laboratory licenses, and difficulty of radioactive waste disposal [16]. For these reasons, the development of new screening methods is necessary for detecting estrogenic EDCs.

Resonance energy transfer is an electrodynamic phenomenon in the relationship between the two molecules, donor (the energy-giving molecule) and acceptor (the energy-receiving molecule), when the distance between the two molecules approaches 1–10 nm and is applied as a powerful system to investigate protein–protein interactions and ligand–receptor interactions [17]. Two main types of RET methods are used: fluorescence resonance energy transfer (FRET) and bioluminescence resonance energy transfer (BRET). In FRET systems, both the donor and the acceptor use fluorophores. A cyan fluorescence protein and the yellow fluorescence protein are the most used FRET pair [18]. The main disadvantage of FRET is the low signal-to-noise ratio caused by the spectral crosstalk between the two fluorophores as well as photobleaching of the acceptor [19]. The BRET method is based on the same principle but replaces the donor with a bioluminescent protein [20]. The bioluminescent donor used in BRET by catalytic oxidation of the substrate transfers energy to the acceptor, which, in turn, emits a fluorescent signal from an acceptor [21]. In comparison to FRET, BRET is much easier to quantify and can obtain enough signal-to-noise at the concentration of a donor forty times lower [22].

The objective of this study was to analyze the binding affinity of EDCs using the BRET technique. To this end, we developed a human cell-based dimerization assay that provides information about EDCs binding to ER, as well as the background for developing an in vitro dimerization assay.

## 2. Materials and Methods

### 2.1. Chemicals

Chemicals 17β-Estradiol (E2), 17α-estradiol, corticosterone, diethylhexyl phthalate, bisphenol A, and 4-nonylphenol were obtained from Wako Chemicals (Richmond, VA, USA), Sigma–Aldrich (St. Louis, MO, USA), and Tokyo Chemical Industry (TCI, Tokyo, Japan). All chemicals were dissolved in DMSO.

### 2.2. Cell Culture

The human embryonic kidney (HEK293) cell line was obtained from the American Type Culture Collection (ATCC, Manassas, VA, USA). HEK293 cells were routinely maintained in minimum essential medium (MEM; Thermo Fisher Scientific, Waltham, MA, USA) supplemented with 10% fetal bovine serum (FBS; Atlas biologicals, Fort Collins, CO, USA) and 50 U/mL penicillin–streptomycin (Gibco, Thermo Fisher Scientific) at 37 °C in a 5% CO_2_ humidified incubator. Before the BRET-based dimerization assays, the cells medium was changed to Opti-MEM™ I Reduced Serum Medium, no phenol red (Opti-MEM; Thermo Fisher Scientific) supplemented with 4% charcoal-stripped FBS (Access Biologicals, Vista, CA, USA), 25 μg/mL Hygromycin B Gold (InvivoGen, San Diego, CA, USA), and 400 μg/mL G418 (InvivoGen), because using charcoal-stripped FBS removes the sex steroids from FBS.

### 2.3. Construction and Preparation of Vector Containing the Estrogen Receptor-Alpha

NanoLuc luciferase (Nluc) fusion protein construction, expression, and purification were performed using the pFN31A Nluc CMV-Hygro Flexi^®^ Vector (6308 bp) for the amino-terminal and the pFC32A Nluc CMV-Hygro Flexi^®^ Vector (6299 bp) for the carboxyl-terminal (Promega, Madison, WI, USA). The HaloTag (HT) fusion protein was constructed using the pFN28A HaloTag^®^ CMV-neo Flexi^®^ Vector (6524 bp) for the amino-terminal and the pFC27A HaloTag^®^ CMV-neo Flexi^®^ Vector (6524 bp) for the carboxyl-terminal, respectively (Promega). ESR1-HaloTag^®^ human ORF in pFN21A (human ERα clone) was purchased from Promega. It was obtained from the human ERα DNA sequence using the restriction-enzyme pair Sgfl/Pmel. The digested DNA fragment was inserted into the Nluc and the HT vectors, respectively. The construction of the fusion vectors was confirmed by agarose gel electrophoresis.

### 2.4. Transient Transfection to HEK293 Cells

The prepared plasmid DNA was transformed into *Escherichia coli* DH5a (Invitrogen, CA, USA) for amplification. Briefly, 2 μL of plasmid DNA was added to 40 μL of *E. coli* and allowed to incubate on ice for 40 min. The transformed cells were then spread on a Luria-Bertani (LB) agar plate supplemented with 50 μg/mL ampicillin. The plasmid was isolated from *E. coli* using QIAprep Spin miniprep kit (Qiagen, Hilden, Germany) according to the manufacture’s recommendations. HEK293 cells were maintained, as described in Section 2.2, and the transfection of the ERα fusion vectors with Nluc and HT was performed in MEM supplemented with 10% FBS and 50 U/mL penicillin-streptomycin. The cells were then seeded in a 6-well plate at a density of 5 × 10^5^; cells per well. The plate with the seeded cells was incubated at 37 °C in a 5% CO_2_ incubator. After 24 h, transient transfection was performed with Lipofectamine™ 3000 Transfection Reagent (Invitrogen) according to the manufacturer’s recommended protocol. To identify the optimal donor–acceptor combination and ratio, lipofectamine 3000 was transiently co-transfect HEK 293 cells with four donor–acceptor combinations (Nluc-ERα + HT- ERα, Nluc-ERα + ERα-HT, ERα-Nluc + HT-ERα, and ERα-Nluc + ERα-HT) at four different donor–acceptor ratios (625 ng:625 ng, 62.5 ng:625 ng, 6.25 ng:625 ng, 0.625 ng:625 ng). These transient transfections were performed according to the technical manual of the NanoBRET^TM^ Protein:Protein Interaction System (Promega).

### 2.5. Bioluminescence Resonance Energy Transfer-Based Dimerization Assay

The BRET signal was measured using a NanoBRET™ Nano-Glo^®^ Detection System (Promega) according to the manufacturer’s guidelines. The medium was removed from the culture dish of transfected cells and washed once with Dulbecco’s phosphate-buffered saline (Gibco). Then, 500 μL of 0.25% trypsin–EDTA (Gibco) was added to separate the cells from the bottom of the culture dish. An equal volume of assay medium was added to Opti-MEM containing 4% charcoal-stripped FBS, 25 μg/mL Hygromycin B Gold, and 400 μg/mL G418 after centrifugation at 12,000× *g* for 5 min. The cells were seeded in 96 well-plates (2 × 10^5^ cells/mL), and 1 μL/mL HaloTag^®^ NanoBRET™ 618 (Promega) as a ligand (+) or 1 μL/mL DMSO as a ligand (−) was added, followed by incubation at 37 °C for 60 min in 5% CO_2_ incubator.

Afterward, the cells were treated with E2 (final concentration of cells, 10^−7^–10^™12^ M) as a reference control, 17α-estradiol (final concentration of cells, 10^−6^–10^−11^ M) as a positive control, corticosterone (final concentration of cells, 10^−6^–10^−11^ M) as a negative control, and DMSO (final concentration of cells, 0.1%) as the vehicle control (VC). The cells were then incubated for 24 h. The NanoBRET^TM^ Nano-Glo^®^ Substrate (Promega) was diluted 100-fold with Opti-MEM and was added to each well after 24 h. Donor emission (460 nm) and acceptor emission (618 nm) signals were measured within 10 min using the GloMax^®^ Discover System (Promega).

### 2.6. Data Acquisition and Analyses

The BRET signal was measured using the GloMax^®^ Discover System (Promega). The result was calculated as follows in Equation (1):(1)$$\mathrm{Raw}\;\mathrm{BRET}\;\mathrm{unit}=\frac{\mathrm{Acceptor}\;\mathrm{emission}\;\mathrm{value}\;\left(618\;\mathrm{nm}\right)}{\mathrm{Donor}\;\mathrm{emission}\;\mathrm{value}\;\left(460\;\mathrm{nm}\right)}$$

The signal was corrected for the background of donor, as shown in Equation (2):(2)Corrected BRET unit=Raw BRET unit in the presence of HaloTag ligand – Mean of raw BRET unit in the absence of HaloTag ligand

Fold induction was calculated as a measured of change between each of the chemical treatment groups and the VC in Equation (3).
(3)$$\mathrm{Fold}\;\mathrm{Induction}=\frac{\mathrm{Corrected}\;\mathrm{BRET}\;\mathrm{unit}\;\mathrm{of}\;\mathrm{the}\;\mathrm{chemical}\;\mathrm{treatment}\;\mathrm{group}}{\mathrm{Mean}\;\mathrm{corrected}\;\mathrm{BRET}\;\mathrm{unit}\;\mathrm{of}\;\mathrm{VC}}$$

The fold induction corresponding to the reference control (E2, 10^−9^ M) value needed to be at least 2.5-fold higher than the mean VC (0.1% DMSO) (=1) for vector combination ratio and position test in BRET assay. The mean difference among experimental E2 or chemical treatment groups was analyzed for statistical significance difference using Sigmaplot ver. 14.0 (Systat Software Inc., San Jose, CA, USA) by one-way analysis of variance (ANOVA), post-hoc Tukey’s test, and Duncan’s test. Statistical significance (*p*-value) was verified within 5%. All data represent the average value from three wells in each experiment and are expressed as mean ± standard deviation for three independent experiments on different days.

## 3. Results

### 3.1. Design and Construction of NanoLuc Luciferase and HaloTag Fused with Estrogen Receptor-Alpha

Vectors encoding all possible combinations of N- (Nluc–ERα, HT–ERα) and C-terminal (ERα–Nluc, ERα–HT) fusion between Nluc and HT with ERα were constructed for testing using the BRET-based dimerization assay. The schematic diagrams are depicted in Figure 1. The vectors had the ampicillin resistance gene, and thus colonies generated on LB plates containing ampicillin were cultured to extract plasmid DNA, which was analyzed for the ERα band by agarose gel separation using SgfI and PmeI or EcoICRI restriction enzymes. Therefore, it was confirmed that the fusion of Nluc and HT at the N- or the C-terminal of ERα was in agreement with the prediction (Figure 2).

### 3.2. Bioluminescence Resonance Energy Transfer Signal by Transient Transfection

To determine the optimized condition for the BRET signal, the combination ratio and the positions of Nluc (as the donor) and HT (as the acceptor) at the N- or the C-terminal of ERα were confirmed. The BRET signal assay was conducted by treating co-transfected HEK293 cells with 10^−9^ M E2 to select the condition of highest fold induction. First, the four types of vector combinations of ERα fused with Nluc and HT were co-transfected into HEK293 cells in the ratios 1:1, 1:10, 1:100, and 1:1000, respectively. Among the combination position of Nluc and HT with ERα, the combination of Nluc and HT fused at the N-terminal of ERα showed significantly increased fold induction compared to other vector combination position (*p* < 0.001) (Figure 3).

Next, the ratios 1:1, 1:10, 1:100, and 1:000, respectively, were compared to determine the optimal ratio of Nluc and HT fused at the N-terminal of ERα on transient transfection into HEK293 cells (Figure 4). The 1:1 ratio was significantly different compared to 1:1000 (*p* < 0.05) and 1:100 (*p* < 0.001), respectively. These results suggested that the optimal combination ratio and position for Nluc and HT vectors to use in the BRET-based ERα dimerization assay were 1:1 and the N-terminal of ERα.

### 3.3. Bioluminescence Resonance Energy Transfer-Based Estrogen Receptor-Alpha Dimerization Assay

Based on the result of optimal combination by transient transfection into HEK293 cells, Nluc and HT fused at the N-terminal of ERα in the ratio 1:1, the feasibility of the application of the BRET technique for ERα dimerization assay was confirmed through the BRET signal. he BRET-based ERα dimerization assay was tested using three chemicals, including E2 and 17α-estradiol as positive chemicals that bind to ERα and corticosterone as a negative chemical that does not bind to ERα. Those chemicals are recommended by the Interagency Coordinating Committee on the Validation of Alternative Test Methods [23] and OECD TG 493 [15]. HEK293 cells were treated with the test chemicals serially diluted to final concentrations from 10^−7^ to 10^−12^ M (for E2) and from 10^−6^ to 10^−11^ M (for 17α-estradiol and corticosterone, respectively). Compared with the VC, the BRET signal for E2 was approximately 2.7 times higher at 10^−9^ M (Figure 5A), and 17α-estradiol was approximate 2.7 times higher at 10^−8^ M, respectively (Figure 5B). These chemicals showed a dose-dependent sigmoidal curve response for ERα. Therefore, E2 and 17α-estradiol were suggested as positive chemicals for ERα. However, corticosterone showed a comparable response to the VC (Figure 5C), which suggests that it is a negative chemical for ERα.

In addition, the BRET-based ERα dimerization assay was conducted for diethylhexyl phthalate, bisphenol A, and 4-nonylphenol, the representative chemicals known as estrogenic endocrine-disrupting chemicals. Compared with the VC, the BRET signals for diethylhexyl phthalate, bisphenol A, and 4-nonylphenol showed a dose-dependent sigmoidal curve for ERα. Among them, diethylhexyl phthalate and 4-nonylphenol showed approximately 2.3 times higher induction fold (Figure 5D,F), while bisphenol A was approximately 5.7 times higher at 10^-5^ M (Figure 5E). These results suggested the possibility that previously known estrogenic chemicals could also show endocrine-disrupting activity in the form of dimerization activity compared to E2.

## 4. Discussion

EDCs may alter the endocrine system in humans and animals by interfering, mimicking, and blocking production, function, and metabolism of hormones [24]. EDCs can be found in food, packaging materials, cosmetics, consumer products, and many other products. Exposure to EDCs can cause metabolic diseases such as carbohydrate, protein, and lipid metabolic disorders, homeostasis dysfunction, hormone imbalance, and hormone-related cancers [25,26]. Therefore, the development of appropriate tests and assessment methods for EDCs is necessary.

EDCs exert adverse effects on the endocrine system, and some EDCs interact with hormone signaling in cells via nuclear hormone receptors, including ERs, androgen receptors, progesterone receptors, thyroid hormone receptors, and retinoid receptors [4,27]. Among them, estrogenic EDCs interfere with the estrogen signaling pathway by interacting with ERs and disrupt the estrogen function in both in vivo and in vitro model systems [28].

Various kinds of in vitro methods for detecting estrogenic EDCs based on the estrogen action were already developed, such as ligand-binding assay, reporter gene assay, transcription assay, and signaling pathway analysis [13]. Among these methods, ligand-binding assays in which the ligand interacts directly with ERs provide a useful detection strategy for estrogenic EDCs [29]. In particular, the radioligand-binding assay in which a radiolabeled ligand binds to the ERα was recommended in the OECD TG 493 [15]. This approach provides outstanding high-throughput chemical screening and testing capability but has spatial and temporal difficulties associated with data acquisition [30]. Alternatively, the BRET assay used in this study overcame some drawbacks of the radioligand- binding assay. BRET-based dimerization assay is suitable for ligand–receptor binding assay and immunological assays of monitoring protein–protein interactions because of a large noise ratio and high sensitivity [20,31]. In addition, the BRET assay can confirm the structural change in the receptor upon binding of the ligand [32]. In the case of ERα, homodimerization occurs when the ligand binds to the ERα [33], and the distance between both ERα decreases so that a BRET signal can be measured. Hence, we consider that the BRET assay could be used to improve existing methods and applied for detecting estrogenic EDCs.

Firefly luciferase (Fluc; 61 kDa, derived from the firefly *Photinus pyralis*) or renilla luciferase (Rluc; 36 kDa, derived from the sea pansy *Renilla reniformis*) are commonly used donors. However, these donors have limitations of large protein size and low stability [34]. The recently added Nluc (19 kDa), derived from the deep sea shrimp *Oplophorus gracilirostris*, can improve the performance of the BRET assay because of its smaller protein size compared with that of existing donors [35]. Furthermore, Nluc is more stable and demonstrates a higher luminescence intensity than Fluc and Rluc [36]. Additionally, as a long-wavelength fluorophore, HT is preferred for the BRET assay because it improves the spectral separation from donor luciferase and enables the rapid evaluation of various fluorophores [37]. Therefore, the NanoBRET system uses Nluc and HT to maintain the activity of Nluc and has higher efficiency than the conventional BRET assay [38].

In this study, we demonstrated the development of an ERα dimerization assay based on the NanoBRET system with Nluc and HT. Before using the NanoBRET system with donor Nluc and acceptor HT, a fusion vector containing ER must be established. The two different isoforms of the ER (ERα and ERβ) both have the estrogen function in humans [39]. ERα is widely expressed and regulates various physiological functions, whereas ERβ is involved in some aspects of mating behavior, regulation of ovulation, and immune responses [40]. In addition, OECD TG 493 describes the binding assay using ERα [15]. In this study, the experiments were performed with an ERα isoform.

We studied transient transfection performed in HEK 293 cells after constructing four different vector combinations of Nluc and HT fused with ERα. Then, we identified the optimal combination vectors by measuring the intensity of the BRET signal after treatment of E2, because it is well known that the BRET signal is influenced by the relative distance of the donor and the acceptor within the ER homodimer [41]. In this study, the highest values of fold induction for BRET signal were selected when compared with VC following E2 treatment. As a result, we obtained the increase fold induction when Nluc and HT vectors were fused at the N-terminal of ERα, respectively, and transient transfection into HEK293 cells in a 1:1 ratio. These data suggested that, if Nluc and HT are located at the N-terminal of ERα, the distance between the two ERαs in the homodimer would be more minimized than other combinations, increasing the efficiency of energy transferred from Nluc to HT, which, in turn, would enhance the BRET signal. Therefore, by establishing the optimal condition for the ERα dimerization assay based on the NanoBRET system, it enhances the performance of the dimerization assay for detecting estrogenic EDCs using the in vitro model.

After establishing the optimal combination vector, we investigated whether using the ERα dimerization assay based on the NanoBRET system is better than using a conventional binding assay. We tested the dimerization ability of chemicals with ERα using the NanoBRET system by selecting the chemicals E2, 17α-estradiol, and corticosterone based on ICCVAM [23] and OECD TG 493 [15]. In this study, E2 and 17α-estradiol showed a dose-dependent response for ERα (Figure 5A and Figure 4B), indicating that these are positive chemicals for ERα, which is consistent with ICCVAM and OECD TG 493. Besides, E2 and 17α-estradiol bound to the ERα to activate estrogen signaling by estrogen responsive elements regulation [28,42]. Accordingly, this finding indicated that these chemicals were bound to ERα to form a homodimer so that the energy of the Nluc donor was transferred to the HT acceptor, which, in turn, emitted fluorescence, enabling measurement of the BRET signal. Contrastingly, corticosterone did not show a dose-dependent response for ERα (Figure 5C). In other words, energy transfer between Nluc and HT by corticosterone did not occur, which confirmed that corticosterone did not induce the formation of ERα homodimers. Therefore, corticosterone is classified as a negative chemical for ERα. Moreover, although this chemical is classified as a presumed negative chemical for ERα in the ICCVAM, the results of this study showed definitively that negative chemicals could be confirmed through the BRET signal, suggesting that the NanoBRET system is more accurate compared with conventional binding assays. Consequently, Nluc and HT can replace the radioligand in the conventional binding assay for ERα that is used to evaluate estrogenic EDCs. Moreover, the novel ERα dimerization assay using the NanoBRET system is able to detect estrogenic EDCs more sensitively and precisely than existing binding assays.

## 5. Conclusions

We established a novel dimerization assay using the NanoBRET system to detect estrogenic EDC-mediated dimerization of ERα. In order to optimize the NanoBRET system, Nluc and HT were selected with a ratio 1:1 at the N-terminal of Erα after vector construction. The combination of Nluc and HT has the advantage that higher efficiency can be maintained relative to other conventional combinations. To measure the BRET signal for ERα, the treating substances were selected with reference to OECD TG 493 and ICCVAM. The results were consistent with both references that conducted the dimerization assay for ERα when using the established NanoBRET system. Moreover, the NanoBRET system can provide accurate substance information. Our results suggest that estrogenic EDCs affecting the estrogen function in vivo can be detected and evaluated by the NanoBRET system more efficiently and accurately. Therefore, this dimerization assay could be used to determine the properties of estrogenic EDCs in foods or substances present in the surrounding environment. Further research is needed to validate our study showing that the BRET signal was confirmed from other EDCs to increase accuracy for the ERα dimerization assay based on the NanoBRET system.

## Figures and Tables

**Figure 1 ijerph-18-08875-f001:**
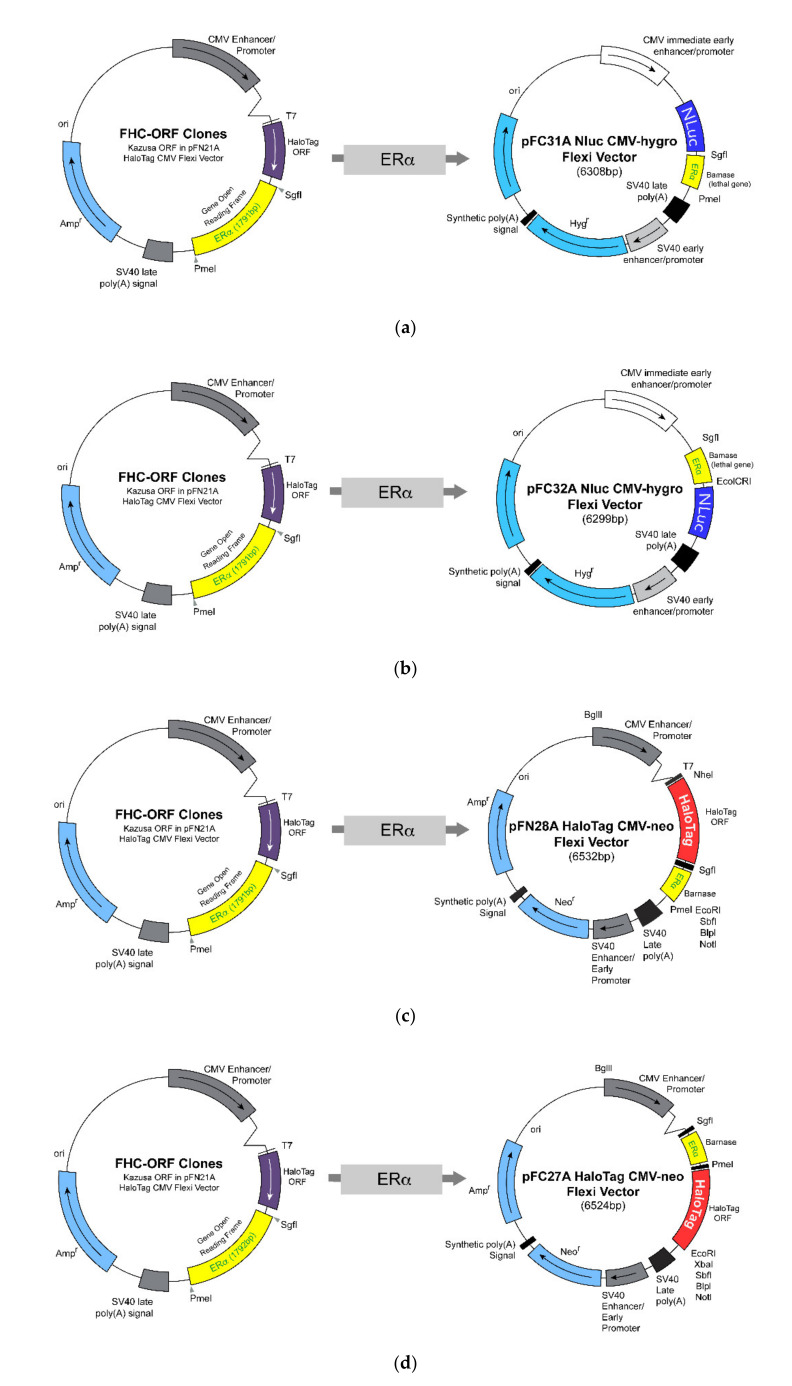
A description of estrogen receptor-alpha (ERα) combined with NanoLuc luciferase (Nluc) and HaloTag (HT). From the top: (**a**) Nluc vector combined at N-terminal of ERα (Nluc–ERα), (**b**) Nluc vector combined at C-terminal of ERα (ERα–Nluc), (**c**) HT vector combined at N-terminal of ERα (HT–ERα), (**d**) HT vector combined at C-terminal of ERα (ERα–HT).

**Figure 2 ijerph-18-08875-f002:**
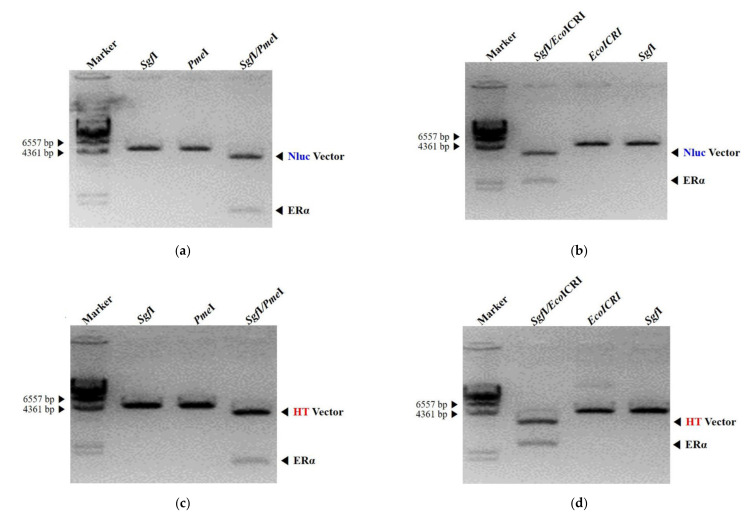
Estrogen receptor-alpha (ERα) fused with NanoLuc luciferase (Nluc) and HaloTag (HT) vectors and analyzed by agarose gel electrophoresis. (**a**) Nluc–ERα: marker, single restriction-enzyme digestion by SgfI, single restriction-enzyme digestion by PmeI, double restriction-enzyme digestion by SgfI/PmeI, respectively. (**b**) ERα–Nluc: marker, double restriction-enzyme digestion by SgfI/EcoICRI, single restriction-enzyme digestion by EcoICRI, single restriction-enzyme digestion by SgfI, respectively. (**c**) HT–ERα: marker, single restriction-enzyme digestion by SgfI, single restriction-enzyme digestion by PmeI, double restriction-enzyme digestion by SgfI, respectively. (**d**) ERα–HT: marker, double restriction-enzyme digestion by SgfI/EcoICRI, single restriction-enzyme digestion by EcoICRI, single restriction-enzyme digestion by SgfI, respectively.

**Figure 3 ijerph-18-08875-f003:**
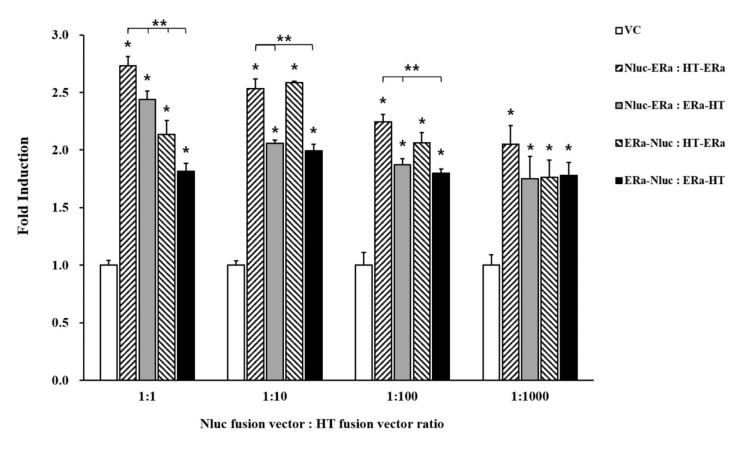
Bioluminescence resonance energy transfer (BRET) signal of the vector combination by transient transfection. BRET signal determined using 10^−9^ M E2. All data results are expressed as means ± SD (*n* = 3). * VC/vector combination ratio group (*p* < 0.001) ** Nluc-ERα:HT-ERα/other vector combination position group (*p* < 0.001).

**Figure 4 ijerph-18-08875-f004:**
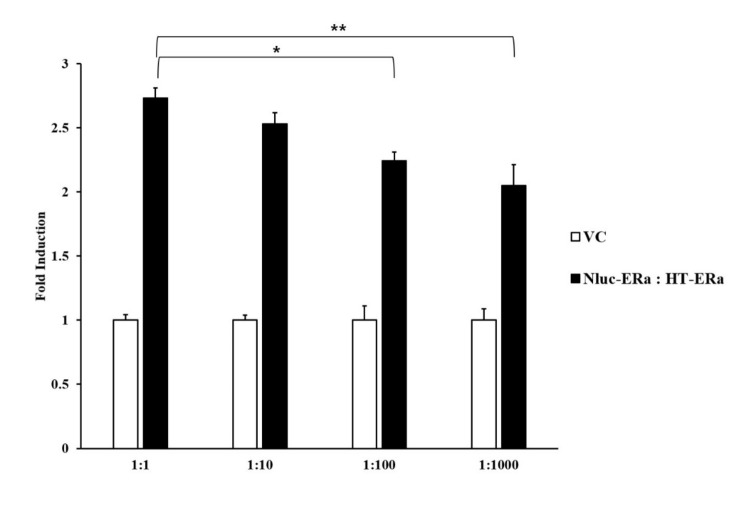
Bioluminescence resonance energy transfer (BRET) signal of the combination 1:1 ratio vector by transient transfection. BRET signal was determined using 10−9 M E2. All data are expressed as means ± SD (n = 3). * 1:1 ratio vector of Nluc-ERα:HT-ERα/1:100 ratio vector of Nluc-ERα:HT-ERα (*p* < 0.05) ** 1:1 ratio vector of Nluc-ERα:HT-ERα/1:1000 ratio vector of Nluc-ERα:HT-ERα (*p* < 0.001).

**Figure 5 ijerph-18-08875-f005:**
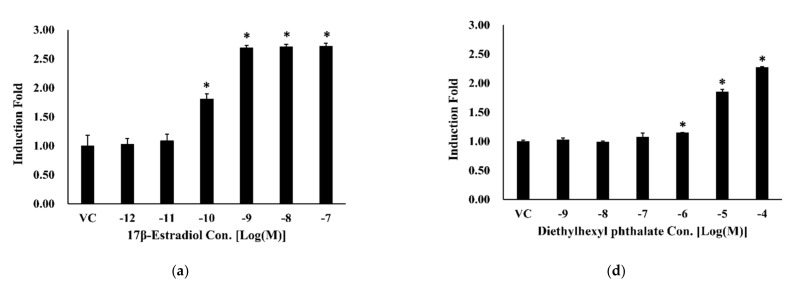
Bioluminescence resonance energy transfer (BRET) signal induced by treating HEK293 cells with (**a**) 10^−7^ to 10^−12^ M E2, (**b**) 10^−6^ to 10^−11^ M 17α-estradiol, (**c**) 10^−6^ to 10^−11^ M corticosterone, (**d**) 10^−4^ to 10^−9^ M diethylhexyl phthalate, (**e**) 10^−5^ to 10^−10^ M bisphenol A, (**f**) 10^−5^ to 10^−10^ M 4-nonylphenol. All data are expressed as means ± SD (*n* = 3). * 1:1 ratio vector of Nluc-ERα:HT-ERα/1:100 ratio vector of Nluc-ERα:HT-ERα (*p* < 0.05).

## Data Availability

The data that support the findings of this study are available from the corresponding author upon reasonable request.

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
