# Peer review of "Development of a Human Estrogen Receptor Dimerization Assay for the Estrogenic Endocrine-Disrupting Chemicals Using Bioluminescence Resonance Energy Transfer"

_ijerph, 2021, doi:10.3390/ijerph18168875_

Round 1
Reviewer 1 Report
This study is tired to establish a new bioassay for detecting estrogenic activities. This work is valuable and innovative. According to the results of this study, it seemed as the preliminary study and I only found the E2 standard tests in this manuscript. For consideration of aim and scope of IJERPH. the current result might be not suitable for publication in IJERPH. Based on application of this bioassay on the environmental investigation or biomonitoring, I recommend to improve the sensitivity and resolution for your bioassay as well as to test the estrogenic chemicals like DEHP, BPA, NP before you resubmit it. Furthermore, it is appreciated to test the environmental samples or biosamples to validate application of your bioassay. I also suggest the authors to reference the the studies for ER-CALUX or T47DKbLuc assay which are successful for detecting estrogenic activities in environmental science.
Author Response
Response to Reviewer 1 Comments
Point 1: This study is tired to establish a new bioassay for detecting estrogenic activities. This work is valuable and innovative. According to the results of this study, it seemed as the preliminary study and I only found the E2 standard tests in this manuscript. For consideration of aim and scope of IJERPH. the current result might be not suitable for publication in IJERPH. Based on application of this bioassay on the environmental investigation or biomonitoring, I recommend to improve the sensitivity and resolution for your bioassay as well as to test the estrogenic chemicals like DEHP, BPA, NP before you resubmit it. Furthermore, it is appreciated to test the environmental samples or biosamples to validate application of your bioassay. I also suggest the authors to reference the the studies for ER-CALUX or T47DKbLuc assay which are successful for detecting estrogenic activities in environmental science.
Response 1: Before responding to the comments you provided, I am very grateful for your review of our results. Also, I would like to express my appreciation for your interest independently of the decision.
Based on the application of this bioassay on the environmental investigation or biomonitoring, environmental samples are compounds of various chemicals, so to evaluate their risks, the composition of each chemical is verified through various analytical analysis methods. Afterward, environmental samples will be evaluated according to regulation corresponding to each chemical.
Our study aims to effectively regulate estrogenic endocrine-disrupting chemicals by providing a new in vitro assay for detecting estrogenic endocrine-disrupting chemicals for regulatory decision of chemicals in environmental samples.
Therefore, rather than directly applying environmental samples to tests based on human cell-based ERα dimerization assay, we would like to evaluate novel or existing chemicals using a BRET-based ERα dimerization assay for effective risk assessment for estrogenic endocrine-disrupting chemicals.
We think this is also appropriate for the aim and scope of IJERPH, " Risk Assessment, Toxicology and Public Health
However, we think it is necessary to evaluate chemicals such as DEHP, BPA, and NP, which are often problematic as estrogenic endocrine-disrupting chemicals in the environment, using a BRET-based ERα dimerization assay. So we added the results to Figure 5. Thank you very much for your comments.
Reviewer 2 Report
Authors developed a human cell-based dimerization assay to detect Rα dimerization using BRETv technology. The assay was optimized by evaluation of the binding affinity of 17β-estradiol (E2), 17α-estradiol, and corticosterone. Although this ttechnique was extensively used in the past, this approach will positively impact the field.
Some concerns:
Line 49. Delete a
Lines 65 to 68
This sentence should be re-written. Resonance energy transfer, per se, is not a method of measuring light, instead is a phenomenon where resonance energy is transferred from a donor to an acceptor and could be used as the basis for a method.
Line 80. was instead of is
Line 81. “This dimerization assay will not only” should be replaced by something like “To this end, we developed a human cell-based dimerization assay that will not only provide....”
Line 82. Delete and
Lines 81 to 83. In this paragraph, authors used the word provide 3 times; should be re-written.
Line 108 and 109. These two sentences are not clear.
Lines 111-112. It would be interesting to see the agarose gels. Are all vectors expressing at a similar level?
Figure 1. Impossible to see vector’s schematic representations, figure should be re-sized.
Line 117. Sentence states: “The plasmid was isolated from E. coli according to the manufacture’s recommendations”, although no purification kit or manufacturer is mentioned.
Line 251. “the distance between both ERα becomes short…” should be replaced by “the distance between both Erα decreases…” or “the distance between both Erα gets shorter”
Line 301. Sentence is not clear.
In other words, the BRET signal is not confirmed because by not binding with 301
ERα, corticosterone is not inducing the formation of an ER homodimer, and the energy 302 transfer between Nluc and HT does not occur.
Other considerations:
It would be helpful to mention the ug of each vector used to co-transfect cells.
Line 198 to 200. Authors state:
“These results suggested that the optimal combi- 198
nation ratio and position for Nluc and HT vectors to use in the BRET-based ERα dimeri- 199
zation assay were 1:1 and the N-terminal of ERα. 200”
Considering the formation of ERα homodimers, competition between Nluc and HT (Nluc/Nluc and HT/HT homodimers could be formed), wasn’t 1:1 ratio the expected optimal ratio? This might have to be discuss.
Author Response
Response to Reviewer 2 Comments
Comments and Suggestions for Authors
Authors developed a human cell-based dimerization assay to detect Rα dimerization using BRET technology. The assay was optimized by evaluation of the binding affinity of 17β-estradiol (E2), 17α-estradiol, and corticosterone. Although this ttechnique was extensively used in the past, this approach will positively impact the field.
Response: Before responding to the comments you provided, I am very grateful for your review of our results. Also, I would like to express my appreciation for your interest independently of the decision.
Some concerns:
Point 1: Line 49. Delete a
Response 1: As recommended, we rephrased the sentence.
Point 2: Lines 65 to 68. This sentence should be re-written. Resonance energy transfer, per se, is not a method of measuring light, instead is a phenomenon where resonance energy is transferred from a donor to an acceptor and could be used as the basis for a method.
Response 2: As recommended, we rephrased the sentence.
Lines 65: Resonance energy transfer is an electrodynamic phenomenon in the relationship between the two molecules, donor (the energy-giving molecule) and acceptor (the energy-receiving molecule) when the distance between the two molecules approaches 1–10 nm, and is being applied as a powerful system to investigate protein-protein interactions and ligand-receptor interactions.
Point 3: Line 80. was instead of is
Response 3: As recommended, we rephrased the sentence.
Point 4: Line 81. “This dimerization assay will not only” should be replaced by something like “To this end, we developed a human cell-based dimerization assay that will not only provide....”
Response 4: As recommended, we rephrased the sentence.
Point 5: Line 82. Delete and
Response 5: As recommended, we rephrased the sentence.
Point 6: Lines 81 to 83. In this paragraph, authors used the word provide 3 times; should be re-written.
Response 6: As recommended, we rephrased the sentence.
Line 81: To this end, we developed a human cell-based dimerization assay that will provide information about EDCs binding to ERα, as well as the background for developing an in vitro dimerization assay.
Point 7: Line 108 and 109. These two sentences are not clear.
Response 7: As recommended, we rephrased the sentence.
Line 108: ESR1-HaloTag® human ORF in pFN21A (human ERα clone) was purchased from Promega.
Point 8: Lines 111-112. It would be interesting to see the agarose gels. Are all vectors expressing at a similar level?
Response 8: There was a mistake. We revised the manuscript with the sentence “The construction of the fusion vectors was confirmed by agarose gel electrophoresis.”
Point 9: Figure 1. Impossible to see vector’s schematic representations, figure should be re-sized.
Response 9: As recommended, we replaced the image file of Figure 1.
Point 10: Line 117. Sentence states: “The plasmid was isolated from E. coli according to the manufacture’s recommendations”, although no purification kit or manufacturer is mentioned.
Response 10: As recommended, we rephrased the sentence.
Point 11: Line 251. “the distance between both ERα becomes short…” should be replaced by “the distance between both Erα decreases…” or “the distance between both Erα gets shorter”
Response 11: As recommended, we rephrased the sentence.
Point 12: Line 301. Sentence is not clear.
In other words, the BRET signal is not confirmed because by not binding with ERα, corticosterone is not inducing the formation of an ER homodimer, and the energy transfer between Nluc and HT does not occur.
Response 12: As recommended, we rephrased the sentence.
Line 301: In other words, energy transfer between Nluc and HT by corticosterone did not occur, which confirmed that corticosterone did not induce the formation of ERα homodimers.
Other considerations:
Point 13: It would be helpful to mention the ug of each vector used to co-transfect cells.
Response 13: As recommended, we added the mention.
Point 14: Line 198 to 200. Authors state: “These results suggested that the optimal combination ratio and position for Nluc and HT vectors to use in the BRET-based ERα dimerization assay were 1:1 and the N-terminal of ERα.
Considering the formation of ERα homodimers, competition between Nluc and HT (Nluc/Nluc and HT/HT homodimers could be formed), wasn’t 1:1 ratio the expected optimal ratio? This might have to be discuss.
Response 14: This consideration is thought to be due to insufficient explanation. As mentioned in the manuscript, the method for co-transfection was to follow the manufacturer’s protocol, which led to a lack of explanation.
As shown in Point 13, the manufacturer’s protocol suggests that transfection for the optimal donor/acceptor combination and ratio should be identified in a way that increases the ratio of acceptor, but it is practically impossible to quantitatively use an acceptor expression vector at 1,000 times as much as the donor expression vector in the transfection. So in fact, in co-transmission, we control the donor/acceptor ratio of 1:1, 1:10, 1:100 and 1:1000 by holding the acceptor expression vector throughput and reducing the throughput of the donor expression vector.
The manufacturer's protocol recommends that raw donor values (Nluc) are typically measured at 1,000,000 to 10,000,000 RLU, with or without ligand. This can reduce errors in the measurement of testing methods using the entire NanoBRET system. In order for raw donor values to be measured at 1,000,000 to 10,000,000 RLU, the amount of donor expression vector treated in transfection was more than 50 ng.
Therefore, we concluded that when the donor/acceptor ratio was 1:1, it was most optimized.
Reviewer 3 Report
The paper is well performed and written. All experimental sections were properly carried out and discussed. The conclusions were supported by the data. I suggest the acceptance of the paper pending revisions, particularly:
- in introduction add the following ref. Journal of Chromatography A Volume 1434, Pages 1 - 18, 2016; Instrumentation Science and Technology Volume 40, Issue 2-3, Pages 112 - 137, 2012
- the terms "in vivo" "in vitro" and others must be reported in italics
- clarify better if in this protocol could be present some quenching phenomena that could modify the method response
Author Response
Reviewer 3
The paper is well performed and written. All experimental sections were properly carried out and discussed. The conclusions were supported by the data. I suggest the acceptance of the paper pending revisions, particularly:
Response: Before responding to the comments you provided, I am very grateful for your review of our results. Also, I would like to express my appreciation for your interest independently of the decision.
Point 1: in introduction add the following ref. Journal of Chromatography A Volume 1434, Pages 1 - 18, 2016; Instrumentation Science and Technology Volume 40, Issue 2-3, Pages 112 - 137, 2012
Response 1: The reference 1 you recommended is a paper that can explain well that EDC can be introduced into the food we eat through water in the environment. So we quoted this paper in the introduction. However, reference 2 was not quoted because it was deemed inappropriate for our introduction.
If you recommend additional citations, we will add it after reviewing it.
Point 2: the terms "in vivo" "in vitro" and others must be reported in italics
Response 2: As recommended, we rephrased the sentence.
Point 3: clarify better if in this protocol could be present some quenching phenomena that could modify the method response
Response 3: As recommended, we rephrased the sentence.
Round 2
Reviewer 1 Report
Thanks for your great effort with modifications. I still have the concern on the induction folds for the EDC standard tests. I criticize the minor induction folds by testing EDC standards. For the bioassay system, it is not available for investigation of EDCs in the environment. I strongly recommend the authors to change their title (like preliminary establishment or development)and list the limitations of the present study. Please response my comments in detail and then I will consider to recommend it for publication.
Author Response
Response to Reviewer 1 Comments (Round 2)
Point 1: Thanks for your great effort with modifications. I still have the concern on the induction folds for the EDC standard tests. I criticize the minor induction folds by testing EDC standards. For the bioassay system, it is not available for investigation of EDCs in the environment. I strongly recommend the authors to change their title (like preliminary establishment or development)and list the limitations of the present study. Please response my comments in detail and then I will consider to recommend it for publication.
Response 1: Thank you again for your sincere comments.
We appreciate your comments because what we need to overcome now is the same as your concern. As you commented, we revised the title of the paper and added the limitations of our study to the discussion section (Line 330).
Reviewer 2 Report
Authors have improved the manuscript, some conrcerns left are:
Line 66: PLEASE CONSIDER REPLACING THE EXISTING SENTENCE BY SOMETHING SUCH AS:
Resonance energy transfer, an electrodynamic phenomenon THAT OCCURS WHEN ENERGY IS TRANFERRED between the two molecules, a donor (the energy-giving molecule) and an acceptor (the energy-receiving molecule), when the distance between the two molecules approaches 1–10 nm, has being applied as a powerful system to investigate protein-protein interactions and ligand-receptor interactions [17].
Distance is not the only requirement for a successful RET, donor emission-acceptor excitation spectrum overlapping and spatial orientation, as well. This should be discussed in the paragraph starting at line 66. Authors can check different publication for this, such as:
- Mild, J. et al. Optimization of a Bioluminescence Resonance Energy Transfer-Based Assay for Screening of Trypanosoma cruzi Protein/Protein Interaction Inhibitors. Molecular Biotechnology 60, 369–379 (2018)
- Borroto-Escuela, et al. Bioluminescence Resonance Energy Transfer Methods to Study G Protein-Coupled Receptor–Receptor Tyrosine Kinase Heteroreceptor Complexes, Methods in Cell Biology, Academic Press, Volume 117, 2013,Pages 141-164.
Author Response
Response to Reviewer 2 Comments (Round 2)
Authors have improved the manuscript, some concerns left are:
Response: Thank you again for your sincere comments.
Point 1: Line 66: PLEASE CONSIDER REPLACING THE EXISTING SENTENCE BY SOMETHING SUCH AS:
Resonance energy transfer, an electrodynamic phenomenon THAT OCCURS WHEN ENERGY IS TRANFERRED between the two molecules, a donor (the energy-giving molecule) and an acceptor (the energy-receiving molecule), when the distance between the two molecules approaches 1–10 nm, has being applied as a powerful system to investigate protein-protein interactions and ligand-receptor interactions [17].
Response 1: As recommended, we rephrased the sentence.
Point 2: Distance is not the only requirement for a successful RET, donor emission-acceptor excitation spectrum overlapping and spatial orientation, as well. This should be discussed in the paragraph starting at line 66. Authors can check different publication for this, such as:
- Mild, J. et al. Optimization of a Bioluminescence Resonance Energy Transfer-Based Assay for Screening of Trypanosoma cruzi Protein/Protein Interaction Inhibitors. Molecular Biotechnology 60, 369–379 (2018)
- Borroto-Escuela, et al. Bioluminescence Resonance Energy Transfer Methods to Study G Protein-Coupled Receptor–Receptor Tyrosine Kinase Heteroreceptor Complexes, Methods in Cell Biology, Academic Press, Volume 117, 2013,Pages 141-164.
Response 2: Of course, we agree with you 100%. Distance is not the only requirement for a successful RET. In particular, the donor emission-acceptor excitation spectrum overlapping is one of the key requirements determining a success of the RET. However, we still believe that the meaning of the distance between the bioluminescence or fluorescence in RET is the key concept of energy transfer. Other factors are key parts that need to be coordinated for a “successful” RET.
So our study have identified how bioluminescent and fluorescence should be fused to estrogen receptor (combination) and how ratio of donor and acceptor should be optimized, as you recommended for the reference [Mild, J. et al. Optimization of a Bioluminescence Resonance Energy Transfer-Based Assay for Screening of Trypanosoma cruzi Protein/Protein Interaction Inhibitors. Molecular Biotechnology 60, 369–379 (2018)]. Also, as you recommended for the reference [Borroto-Escuela, et al. Bioluminescence Resonance Energy Transfer Methods to Study G Protein-Coupled Receptor–Receptor Tyrosine Kinase Heteroreceptor Complexes, Methods in Cell Biology, Academic Press, Volume 117, 2013,Pages 141-164] the Discussion section covered the reason for applying the NanoBRET system, the donor emission-acceptor excitation spectrum overlapping (Line 274).
The only reason why the introduction was so simple was to provide intuitive information to readers who are new to the RET system. We are responsible for what you felt because this purpose did not clearly state the sentence of line 66 against our intention. We Just want to say that we also considered various factors in RET and put them in paper.
If you want these parts to be mentioned in the introduction, not in the discussion section, please comment again. we will consider it actively.